



# Evaluation of satellite methods for estimating supraglacial lake depth in southwest Greenland

Laura Melling[1], Amber Leeson[1], Malcolm McMillan[1], Jennifer Maddalena[1], Jade Bowling[1], Emily Glen[1], Louise Sandberg Sørensen[2], Mai Winstrup[2], Rasmus Lørup Arildsen[2]

[1]Lancaster Environment Centre, Lancaster University, Lancaster, LA1 4YQ, United Kingdom
[2]DTU Space, Danmarks Tekniske Universitet, Lyngby, 2800, Denmark

*Correspondence to*: Laura Melling (l.melling@lancaster.ac.uk)

**Abstract.** Supraglacial lakes form on the Greenland ice sheet in the melt season (May to October) when meltwater collects in surface depressions on the ice. Supraglacial lakes can act as a control on ice dynamics since, given a large enough volume of

water and a favourable stress regime, hydrofracture of the lake can occur which enables water transfer from the ice surface to the bedrock where it can lubricate the base. The depth (and thus volume) of these lakes is typically estimated by applying a radiative transfer equation (RTE) to optical satellite imagery. This method can be used at scale across entire ice sheets but is poorly validated due to a paucity of in-situ depth data. Here we intercompare supraglacial lake depth detection by ArcticDEM digital elevation models, ICESat-2 photon refraction, and the RTE applied to Sentinel-2 images across five lakes in southwest

Greenland. We found good agreement between the ArcticDEM and ICESat-2 approaches (Pearson's r = 0.98) but found that the RTE overestimates lake depth by up to 153 % using the green band (543–578 nm) and underestimates lake depth by up to 63 % using the red band (650–680 nm). Parametric uncertainty in the RTE estimates is substantial and is dominated by uncertainty in estimates of reflectance at the lakebed which are derived empirically. Our analysis indicates that calculating depth with the RTE using literature-derived values for the parameters introduces significant uncertainty in the retrieval of

depth information from optical imagery. Uncertainty in lake depth estimates translates into a poor understanding of total lake volume, which could mean that hydrofracture likelihood is under or over-estimated, in turn affecting ice velocity predictions. Further laboratory studies to constrain spectral radiance loss in the water column, and investigation of the potential effects of cryoconite on the estimation of lakebed reflectance could improve the RTE in its current format. However, we also suggest that future work should explore data-driven approaches to deriving lake depth from optical satellite imagery, which may

improve depth estimates and will certainly result in better-constrained uncertainties.

## 1 Introduction

Supraglacial lakes form when meltwater collects in surface depressions on glaciers and ice sheets. On the Greenland ice sheet, lakes form in approximately the same locations each melt season from May to October (Sundal et al., 2009), as their positions are controlled by bedrock topography (Echelmeyer et al., 1991; Krawczynski et al., 2009). Alongside rivers and streams,

supraglacial lakes form a complex hydrological system of water storage and transport on the ice sheet surface. As the melt season progresses, supraglacial lakes grow in size through the accumulation of meltwater. These lakes either drain or refreeze





(Selmes et al., 2013). Drainage can occur slowly over the ice surface through supraglacial channels or rapidly through the ice if the weight of the water is sufficient to drive a crevasse through the full ice thickness to the bed (hydrofracture). Hydrofracture-induced drainage events can occur in as little as two hours (Das et al., 2008). In these events, the water is routed

to the base of the ice sheet where it can cause a hydraulic pressure increase that temporarily lifts the ice off the bed. This process, known as hydrofracturing, can enhance basal sliding and increase ice flow rates (Fitzpatrick et al., 2013; Tedesco et al., 2013; Christoffersen et al., 2018; Tuckett et al., 2019; Maier et al., 2023). Short-term increases in meltwater input cause temporary spikes in water pressure which lead to ice acceleration whereas an increase in mean melt supply does not necessarily cause an increase in ice sheet velocity (Schoof, 2010). Ergo, knowing the volume of water held on the ice sheet at any one

time - and thus the potential for temporary spikes in water pressure through hydrofracture - is important for modelling ice sheet dynamics.

Our understanding of ice sheet behaviour assumes that we have an understanding of meltwater delivery (Zwally et al., 2002; Parizek and Alley, 2004). If the calculated depth of supraglacial lakes is inaccurate, the volume of the lake is inaccurate, thus meaning our calculations of injected meltwater to the ice sheet bed are also inaccurate. As a result, under or overestimating

the volume of meltwater holds consequences for the models on which we base our understanding of ice dynamics.

To understand the amount and rate of meltwater delivery to the ice sheet bed we require spatially and temporally continuous observations of lake volume. Our study area, located in the southwest Greenland ice sheet (Fig. 1), includes the lower Watson River basin (5800 km$^2$). This basin has a meltwater coverage (including supraglacial lakes, streams, and rivers) of 250 km$^2$ (E Glen 2022, personal communication, 22 July), meaning it is not feasible to acquire spatially and temporally continuous lake

volume data from field surveys. Instead, several satellite-based methods can be used to estimate supraglacial lake depths remotely, potentially providing high spatial and temporal coverage. These methods are; physics-based modelling, such as the application of the radiative transfer equation (RTE) proposed by Philpot (1987) to optical satellite imagery (Moussavi et al., 2020); laser altimetry, which is used to measure lake depths directly from photon refraction (Fair et al., 2020); the use of digital elevation models to ascertain lake depth from the underlying ice surface topography (Yang et al., 2019); and empirical models

derived from regression of in-situ depth measurements with remotely sensed data (Pope et al., 2016).

These methods each have known advantages and limitations for deriving lake depths. Physics-based models applied to optical satellite data (e.g. Sentinel-2) provide continuous spatial coverage at high-resolution temporal sampling (i.e. every 5 days), and they can be used at scale. ICESat-2 can directly measure lake depths but is limited to 1-D profiles along discrete satellite tracks which are spatially distant (4.1 km between acquisition beams of neighbouring satellite tracks at 67°N) and have coarse

temporal sampling, inhibiting an assessment of lake dynamics as supraglacial hydrology evolves on sub-daily timescales (Das et al., 2008). ArcticDEM data is even more sporadic in space and time, with periods of months between acquisitions and missing data caused by cloud cover. However, ArcticDEM offers a spatial resolution (2 m) of an order of magnitude higher than Sentinel-2 and thus enables a more detailed assessment of lake bathymetry; for example, to assess whether a lake basin contains open or healed crevasses that may promote lake drainage by hydrofracture. Although empirical models derived

through the regression of in-situ depth measurements with remotely sensed data have been shown to define reasonable lake





depth (Pope et al., 2016), their coefficients are spatially constrained to the area in which the original in-situ measurements were taken and are, therefore, unreasonable to apply on the ice sheet scale. Here, we examine and intercompare the performance of a physics-based model, ICESat-2 laser altimetry and ArcticDEM digital elevation models in determining the depth of a test dataset of supraglacial lakes in the southwest Greenland ice sheet, where Greenlandic supraglacial lakes are extensive and

numerous (Hu et al., 2022).

## 2 Data and Methods

### 2.1 Study region and supraglacial lakes

Our study region (Fig. 1) is located in western Greenland and contains part of the Watson River basin, known for abundant supraglacial lake coverage. This region contains active (repeatedly filling and draining) lakes and is known to respond

dynamically to hydrological perturbations (e.g. Das et al., 2008; Chudley et al., 2019). Five lakes in this region were found to be suitable for depth retrieval from all three of the datasets (with the availability of ICESat-2 data being the main limitation, see Appendix A.1). Each of the five supraglacial lakes are active, are crossed by an ICESat-2 ground track, and have both concurrent optical imagery and a corresponding digital elevation model (DEM) showing an empty lake basin. These five lakes span a range of sizes (0.8 km$^2$–3.1 km$^2$) and appear at a range of elevations (1150–1500 m a.s.l.), as shown in Fig. 1.


**Figure 1: The locations of the five supraglacial lakes in relation to the study region. Contour lines calculated from the ArcticDEM 100 m mosaic are visible on the base map as grey dashed lines. The inset map indicates the location of the study area within Greenland. (1)–(5) show Lake 1 to Lake 5 in detail, where the background is a true colour image acquired on the date shown in Table A1 for each lake. The manually delineated lake outline is given in red and the ICESat-2 transect is given in orange. The ICESat-2 ground tracks were cropped to the lake edges. The base map data is courtesy of Earthstar Geographics via ESRI.**

We apply three different methods to measure the depth of each of our five lakes. These methods are described in detail below.

**2.2 Method 1: Optical satellite imagery**

The RTE was applied to level 1C Sentinel-2 optical satellite imagery (Table A1) which has a spatial resolution of 10 m and a revisit period in this region of about five days (Drusch et al., 2012). We use the RTE (Eq. 1) first presented in Philpot (1987) and apply the equation to both red (0.65–0.68 μm) and green (0.54–0.58 μm) bands. These bands were chosen as they represent the visible part of the electromagnetic spectrum which optical satellite instruments are designed to detect. They have also





previously been used to determine lake depth on the Greenland ice sheet (e.g. Williamson et al., 2018; Moussavi et al., 2020;
Datta and Wouters, 2021).

We chose imagery which was temporally (± ten days) and spatially concurrent with the ICESat-2 data. Following this, we undertook a visual appraisal of satellite imagery approximately 15 days on either side of the ICESat-2 acquisition date to ensure that the lakes had not changed in size during this period, and hence that the optical imagery and ICESat-2 depths were temporally comparable. To ensure the comparative equivalence of our methodologies, we followed previous studies (e.g.
Williamson et al., 2018; Moussavi et al., 2020; Datta and Wouters, 2021) and used Sentinel-2 Level 1C products which are pre-packaged as orthorectified, map-projected imagery of scaled top-of-atmosphere data. We converted these values to unscaled top-of-atmosphere reflectance.

The radiative transfer approach to modelling lake depth is based on the assumptions of the Bouguer-Lambert-Beer Law. Equation (1) gives the formulation of the equation presented by Philpot (1987), written in terms of reflectance and inverted
into the logarithmic form.

$$z = \frac{\ln(A_d - R_\infty) - \ln(R_w - R_\infty)}{g} ,\qquad(1)$$

$A_d$ represents the lake bottom albedo/reflectance, $R_\infty$ indicates the reflectance of optically deep water, $R_w$ is the recorded reflectance of a given pixel, $g$ is the coefficient for spectral radiance loss in the water column, and $z$ represents lake depth in metres. We assume the lake substrate is homogenous, suspended or dissolved particles are minimal, there is no inelastic
scattering or fluorescence, effects of wind are minimal, and lake bottom pixels are parallel to the lake surface, following Sneed and Hamilton (2011). $A_d$, $R_\infty$, and $g$ are all tuneable parameters.

$A_d$ represents the lake bottom albedo which is typically approximated by the reflectance of a ring of pixels around the lake. We take $A_d$ to be the average reflectance value in a 30 m wide ring (three pixels in Sentinel-2, after Moussavi et al. (2020)) around each of the five lakes to provide a unique $A_d$ value for each lake.
$g$ is the coefficient for spectral radiance loss in the water column and accounts for the scattering of both downwelling ($K_d$) and upwelling ($K_u$) light (Eq. 2).

$$g = K_d + K_u ,\qquad(2)$$

$K_d$ is wavelength specific. The minimum value is determined using Eq. (3) (Smith and Baker, 1981),

$$K_d = a_w + \frac{1}{2} b_m^{fw} ,\qquad(3)$$

where $a_w$ is the absorption coefficient for pure water, and $b_m^{fw}$ is the backscattering coefficient for molecular scattering in freshwater.

We averaged the $a_w$ and $b_m^{fw}$ values from Smith and Baker (1981) for both bands and calculated the band-specific $K_d$ values using Eq. (3).





Many laboratory-derived estimates exist of $K_d$ but very few exist of $K_u$ (Philpot, 1989). Other studies have taken $K_u$ to be equal

to $K_d$, and thus $g$ to be $2K_d$ (e.g. Maritorena et al., 1994; Sneed and Hamilton, 2007), but $K_u$ must be larger than $K_d$ because the horizontally-biased angular distribution of upwelling photons backscattered by an infinitely-thin layer is known to be more rapidly attenuated than the downwelling photon flux (Kirk, 1989). In fact, experimental studies suggest that $g$ could be as high as $3.5K_d$ (Kirk, 1989), with some studies suggesting a higher $g$ value leads to more accurate lake depths (Brodský et al., 2022). Here, we therefore use an average of this range and take $g = 2.75K_d$.

$R_\infty$ represents reflection from the water column and is commonly taken as the reflectance of optically deep, clear and still water i.e. where it is reasonable to assume there is no bed reflectance or sediment contamination (Sneed and Hamilton, 2007). To counter the potential impacts of choosing a value of $R_\infty$ that had been incorrectly identified as uncontaminated optically deep water (e.g. the pixel contained sediment traces or was dark as a result of sensor-related scanning issues), we manually appraised the darkest pixels from each lake's $R_\infty$ scene (Appendix A.2) for anomalous data, and we removed pixels which exhibited these

values. After this, we identified the ten darkest, uncontaminated pixels in each scene and averaged them to produce the unique $R_\infty$ values for each lake scene.

The methods with which we have calculated the $g$ and $R_\infty$ parameter values vary slightly from previous studies (e.g. Sneed and Hamilton, 2007; Pope et al., 2016). Hereon, where we refer to the 'literature values' we mean $g = 2K_d$ (Sneed and Hamilton, 2007), where $K_d$ is as described in Eq. (3), and $R_\infty$ is the reflectance of the darkest pixel in the deep-sea scene (e.g. Georgiou

et al., 2009; Pope et al., 2016).

We calculated the uncertainty of the RTE depths along each ICESat-2 track by iterating through every permutation in the ranges of the tuneable parameters for each lake (Appendix A.3). The standard deviation of these permutations at every depth detection point along the lake's ICESat-2 transect was taken to represent the uncertainty of the depth measurement. Points along the transect are spaced at approximately 0.7 m intervals (1292 points for Lake 1, 1944 points for Lake 2, 1244 points for

Lake 3, 1346 points for Lake 4, and 2138 points for Lake 5). We defined depth detection points in this way to ensure the datasets were sampled at the spatial resolution of ICESat-2, which is our highest-resolution dataset.

### 2.3 Method 2: ArcticDEM

ArcticDEM is an open-access collection of high-resolution DEMs produced by the Polar Geospatial Center. The dataset is assembled from individual stereoscopic DEMs that are derived from pairs of high-resolution optical imagery, acquired by the

WorldView-1, WorldView-2, WorldView-3, and GeoEye-1 satellites (Morin et al., 2016). The DEMs are generated by applying the Surface Extraction from TIN-based Searchspace Minimization (SETSM) software to stereoscopic image pairs (Noh and Howat, 2017). Here, we use the most recent version of ArcticDEM data (version s2s041; release 8). The tile reference numbers of the DEMs used in this study are detailed in Table A1; all available ArcticDEM DEMs of the study region were acquired from the Polar Geospatial Center.

The DEMs used in this study correspond to empty supraglacial lake basins; the bathymetry (and thus potential depth) of an empty lake basin is visible in a DEM, whereas a filled lake basin presents as a flat surface. Due to the sparse temporal sampling





of ArcticDEM, and the need to resolve empty basins, the DEMs are not temporally concurrent with the ICESat-2 and Sentinel-2 data. As a result, the smallest period between the ArcticDEM and ICESat-2 acquisition dates was approximately two months (Lake 4), and the largest period was approximately 11 months (Lake 5) (Table A1). Despite the potential for changes in the
bathymetry of Lake 5 over the 11-month period between acquisition dates, the location and shape of supraglacial lakes are determined by bedrock topography (Echelmeyer et al., 1991), so we assume there should be little change in the bathymetry (see Sect. 3.1 for further details).

To calculate lake depth from a DEM, we need to examine the shape of the basin before or after it has drained. As drained lakes have similar characteristics to perpetually dry surface depressions, we first identified which depressions in the DEMs were
associated with lakes. To identify active lakes in our study region, we followed the approach outlined in Bowling et al. (2019) to stack DEMs and calculated the temporal variance in elevation at every pixel. Whereas Bowling et al. (2019) used this approach to detect subglacial lakes, here we use it to identify supraglacial lakes, hence we tailor the threshold on the elevation variance accordingly. Specifically, we select features where the standard deviation lies in the range of 2-7 m; below this threshold, variation in elevation can arise from artefacts in the DEM; and ICESat-2 depth detection is limited to lakes up to 7
m deep (Fair et al., 2020).

We then deployed this standard deviation range across the study region to find areas of high standard deviation which indicate potentially active lakes. After the active areas were identified, we cross-referenced them with the locations of known supraglacial lakes which met specific characteristics (Appendix A.1) to isolate areas which represented active lakes. This removed potential artefacts from the data which were not caused by lake drainage and refilling.

For each lake, the DEMs were then vertically adjusted to align with the ICESat-2 data. The ends of the ICESat-2 transect represent the very edges of the lakes (i.e. the depth is 0 m), so we identified the DEM elevation value at these points and subtracted the average value from the DEMs. This then gave us an ArcticDEM-derived lake depth, assuming the lake reaches the extent indicated by ICESat-2 and that the ICESat-2 and ArcticDEM data were spatially coregistered. These relative depths were then compared against the depths retrieved from ICESat-2 and the RTE.

**2.4 Method 3: ICESat-2**

ICESat-2 data were used to derive depths delineated along the altimeter tracks which intersected supraglacial lakes. ICESat-2 was launched in 2018 and has a 91-day repeat period, six acquisition beams, and a nominal along-track resolution of 0.7 m (Abdalati et al., 2010), but has non-continuous spatial coverage due to its instrumental and orbital characteristics. At 67°N, for example, the across-track spacing between reference ground tracks (RGTs) is ~10.7 km, with a distance of ~4.1 km between
the right acquisition beam of one RGT and the left acquisition beam of the neighbouring RGT.

We estimate the lake bathymetry of the supraglacial lakes using the ICESat-2 ATLAS ATL03 data product (Table A1), based on the distinct photon returns received from the lake surface (air-water) and bed (water-ice) interfaces. The ATL03 data product provides geolocated photons but does not account for the refraction of photons in the air-water interface resulting from the change in refractive index between the two media. This change in photon speed and paths gives rise to horizontal and vertical



errors in the geolocation record, causing the photon locations to appear deeper in the lake and further off-nadir. We corrected the photon locations using the method described in Parrish et al. (2019).

Our ICESat-2 lake bathymetry estimation algorithm divides the lake into 30 m wide vertical sections. Once divided, we removed photon returns which were outside the 5th to 50th percentile range and also removed photons corresponding to the surface to ensure they did not interfere with lake bathymetry estimation. We invited 10 altimetry experts to manually digitise the lake bathymetry from the ATL03 photon data plots produced by our algorithm using an online digitisation tool (https://apps.automeris.io/wpd/). We took the average of these manual delineations to be the best estimate of lake bathymetry and used the standard deviation of these estimates as an indication of the bathymetry uncertainty.





## 3 Results

### 3.1 Supraglacial lake depths from ArcticDEM, ICESat-2 and the RTE

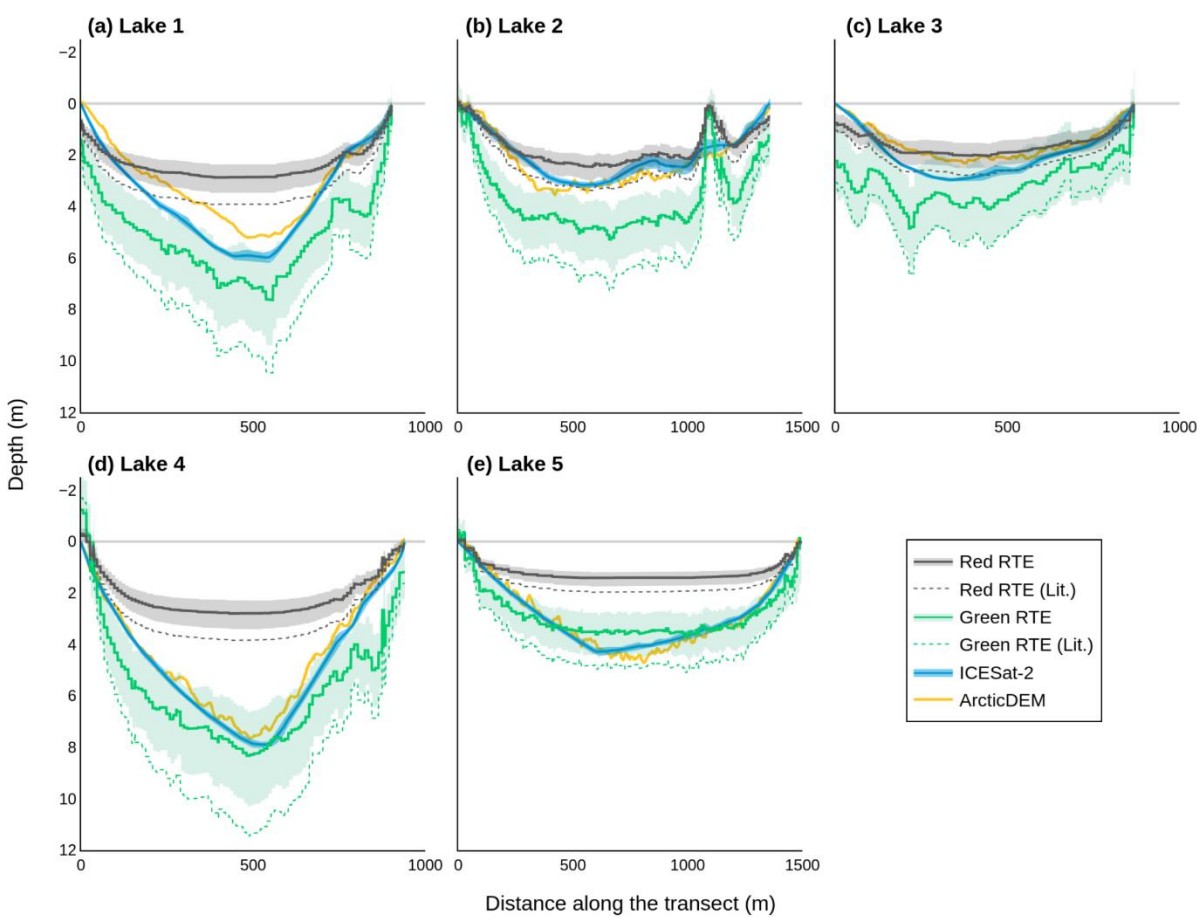

**Figure 2: Supraglacial lake depth from the band-specific RTE, with both literature values and the values used for this study, ArcticDEM, and ICESat-2 along ICESat-2 transects. Depths achieved using the RTE with the literature values ("Red RTE (Lit.)", "Green RTE (Lit.)") are shown for contextual reference. All uncertainties are calculated as one standard deviation. ArcticDEM absolute elevation accuracy is less than five metres in the vertical plane (Noh and Howat, 2015). Lake 2 (b) exhibits a spike in both red band RTE and green band RTE depths at approximately 1200 m along the transect which we attribute to a slight ice covering in the Sentinel-2 imagery. Transect locations are detailed in Fig. 1.**

We calculated depths along the ICESat-2 transects over the five lakes using ArcticDEM, ICESat-2, and the RTE (Fig. 2). The ArcticDEM, red band RTE and green band RTE are sampled approximately every 0.7 m along the transect whereas the ICESat-2 data is sampled at 100 equally spaced points along the transect. We attribute the noise in the RTE transects to differences in spatial resolution (where Sentinel-2 has the coarsest sampling of the three datasets). Here, we choose to evaluate at each sensors' native resolution in keeping with previous studies. However, we note that the application of low pass filters to smooth the optical solutions could be explored in future work.





The red band RTE depths plateau between 1 and 3 m, reaching their deepest depths at 2.87 m, 2.46 m, 2.04 m, 2.81 m, and 1.44 m for lakes 1, 2, 3, 4, and 5 respectively. This plateau typically results in an underestimation of maximum depth. In

contrast, the green band RTE depths show a systematic overestimation compared to ArcticDEM and ICESat-2. The RTE depths are deeper when literature values (Sneed and Hamilton, 2007; Georgiou et al., 2009; Pope et al., 2016) are used as opposed to our parameter values.

The maximum difference between any two methods differs for each lake in both the value and the method pair (Table 1). Here, we disregard the RTE depths retrieved using the literature values as these are shown only for contextual reference. For each

of the five lakes, the red or green band RTE is a component of each of these maximum difference method pairs, which is likely due to the observed under and overestimation of the red band RTE and green band RTE respectively.

Table 1: The method pairings with the maximum depth difference for each of the five lakes.

| Lake number | Method pairing | Maximum depth difference (m) |
|---|---|---|
| 1 | Red band RTE and Green band RTE | 4.77 |
| 2 | Red band RTE and Green band RTE | 2.83 |
| 3 | ArcticDEM and Green band RTE | 3.08 |
| 4 | Red band RTE and Green band RTE | 5.52 |
| 5 | Red band RTE and ArcticDEM | 3.28 |

To explore the agreement between the different methods, we calculated the root mean square difference (RMSD) and Pearson's correlation coefficient for each method pairing (Fig. 3).





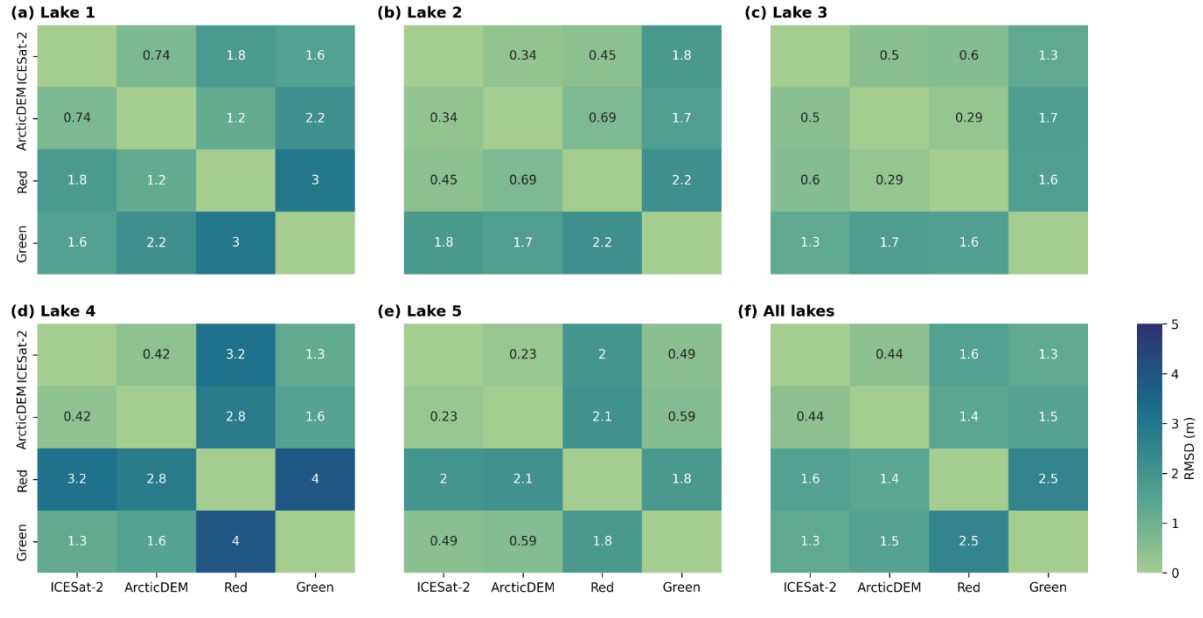

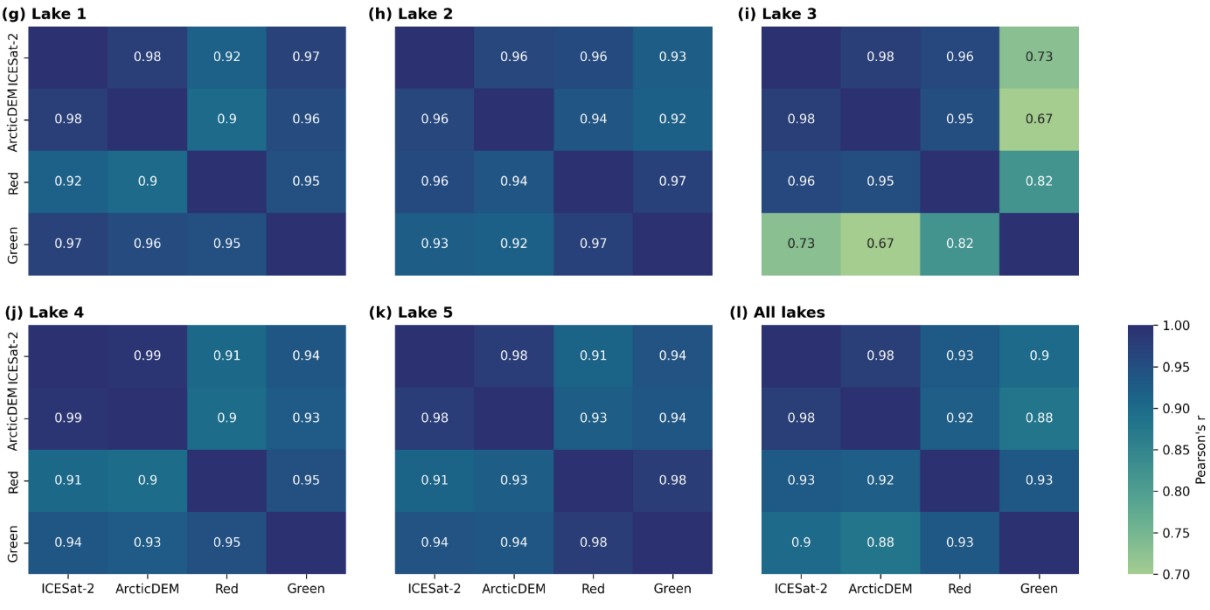

**Figure 3: The root mean square difference (RMSD) and Pearson's correlation coefficients for each paired combination of depths derived from ICESat-2, ArcticDEM, red band RTE and green band RTE. (a)-(e) show the RMSD for Lake 1 to Lake 5, and (f) shows the average RMSD for all five lakes. (g)-(k) show the Pearson's correlation coefficient for Lake 1 to Lake 5, and (l) shows the average Pearson's correlation coefficient for all five lakes.**





From Fig. 3, the method pairing with the lowest RMSD is ICESat-2 and ArcticDEM for all lakes except Lake 3. For Lake 3,
the method pairing with the lowest RMSD is the red band RTE and ArcticDEM. On average, the ICESat-2 and ArcticDEM
pairing has an RMSD of 0.44 m and the method pairing with the highest average RMSD is the red band RTE with the green
band RTE at an RMSD of 2.5 m. Additionally, our results indicate a high degree of agreement between the ArcticDEM depths
and the ICESat-2 depths for Lake 5 (RMSD = 0.23, $r = 0.98$), enhancing our confidence in the lack of bathymetry change over
the 11-month period between data acquisitions.

The Pearson's correlation coefficient of each of the method pairings is significant at $p < 0.001$. However, some of the method
pairings have stronger correlations than others. The method pairing with the strongest correlation for all of the lakes is ICESat-
2 and ArcticDEM with an average $r$ value of 0.98. The method pairing with the weakest correlation is different for each lake.
For Lake 1 and Lake 4, the method pair with the lowest Pearson's $r$ value is the red band RTE and ArcticDEM ($r = 0.90$ for
both). For Lake 2 and Lake 3, the pair with the lowest $r$ value is the green band RTE and ArcticDEM ($r = 0.92$ and 0.67
respectively). For Lake 5, the weakest correlation is for the pairing of the red band RTE and ICESat-2 ($r = 0.91$). The method
pair with the weakest average Pearson's correlation is the green band RTE and ArcticDEM ($r = 0.88$), though this value is
heavily impacted by the results from Lake 3.





## 3.2 ArcticDEM versus RTE: 2D Comparison of supraglacial lake depths

**Figure 4: A comparison of red and green band RTE depths versus ArcticDEM depths in two dimensions. Each column shows results from one of our five study lakes, and each row shows information relating to a different retrieval method. The true colour imagery is from Sentinel-2 (Table A1). ICESat-2 transects are shown in orange on the true colour imagery.**





Next, we extend the 1-D analysis to two dimensions in a comparison between ArcticDEM and the RTE over the entire area of each lake (Fig. 4). Again, we find that the red band RTE depths plateau at depths consistent with Fig. 2. Contrastingly, the

green band RTE depths do not have visible plateau depths for these lakes. Instead, this method again overestimates depths compared to ArcticDEM. Table 2 details the average difference of the green band RTE and red band RTE, in comparison to ArcticDEM.

**Table 2: The average overestimation by the green band RTE and red band RTE depths and volumes when compared to ArcticDEM**
**DEMs for each of the five lakes. All volume estimates are shown to three significant figures.**

| Lake number | Average depth difference (red) (m) | Average volume difference (red) (m$^3$) | Average depth difference (green) (m) | Average volume difference (green) (m$^3$) | Volume estimated by ArcticDEM (m$^3$) |
|---|---|---|---|---|---|
| 1 | -0.06 | -133,000 (-3 %) | +2.05 | +4,230,000 (+106 %) | 4,000,000 |
| 2 | +0.13 | +111,000 (+9 %) | +2.26 | +1,870,000 (+153 %) | 1,230,000 |
| 3 | -0.01 | -10,900 (-1 %) | +1.22 | +997,000 (+89 %) | 1,130,000 |
| 4 | -1.96 | -5,210,000 (-50 %) | +0.94 | +2,500,000 (+24 %) | 10,400,000 |
| 5 | -2.16 | -5,870,000 (-63 %) | +0.52 | +1,420,000 (+15 %) | 9,260,000 |

To further explore the red band RTE depth plateauing effect and identify any noteworthy patterns in the relationship between green-band RTE depths and ArcticDEM depths, we compared the red and green band RTE depths versus the ArcticDEM depths for all ArcticDEM pixels across the five lakes (Fig. 5).





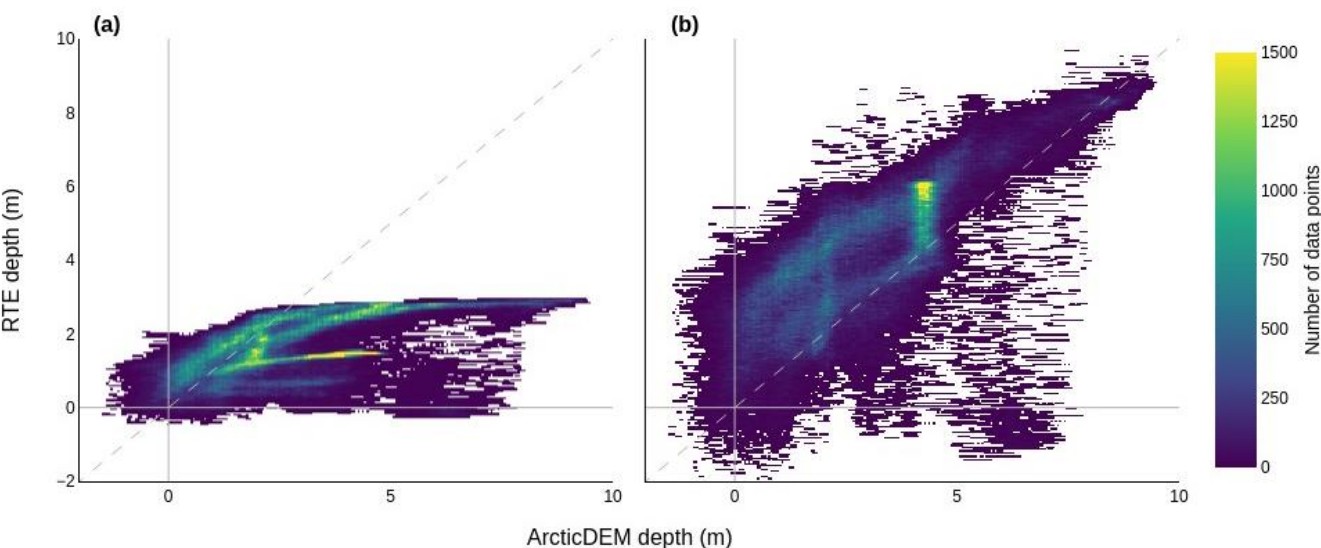


**Figure 5: Density scatter plots of the depths derived from the red (a) and green (b) RTEs versus ArcticDEM depths for every pixel of the five lakes. The dashed lines represent one-to-one agreements between the depth datasets.**

Figure 5 shows the relationship between the depth values of ArcticDEM, the red band RTE, and the green band RTE for every pixel of all five lakes. We find that the red band RTE depth plateauing effect is clearly evident, with each of the lakes having

a different plateau depth as suggested in Fig. 2 and Fig. 4. This variance in red band RTE depth saturation between lakes can be seen in the dense, elongated clusters of the red band RTE cloud, each of which can be attributed to a different lake. We attribute the difference in plateau depth to the varying $A_d$ values of the lakes; with red $A_d$ values of 0.42, 0.46, 0.35, 0.58, and 0.57 for lakes 1, 2, 3, 4, and 5 respectively. Shallower depths (typically towards the lake edges, as seen in Fig. 4) agree relatively well when derived from the red band RTE and the ArcticDEM. However, as the lake gets deeper (towards the centre

in most cases, as seen in Fig. 4), the agreement between the red band RTE depths and the ArcticDEM depths decreases.



## 3.3 RTE sensitivity analysis

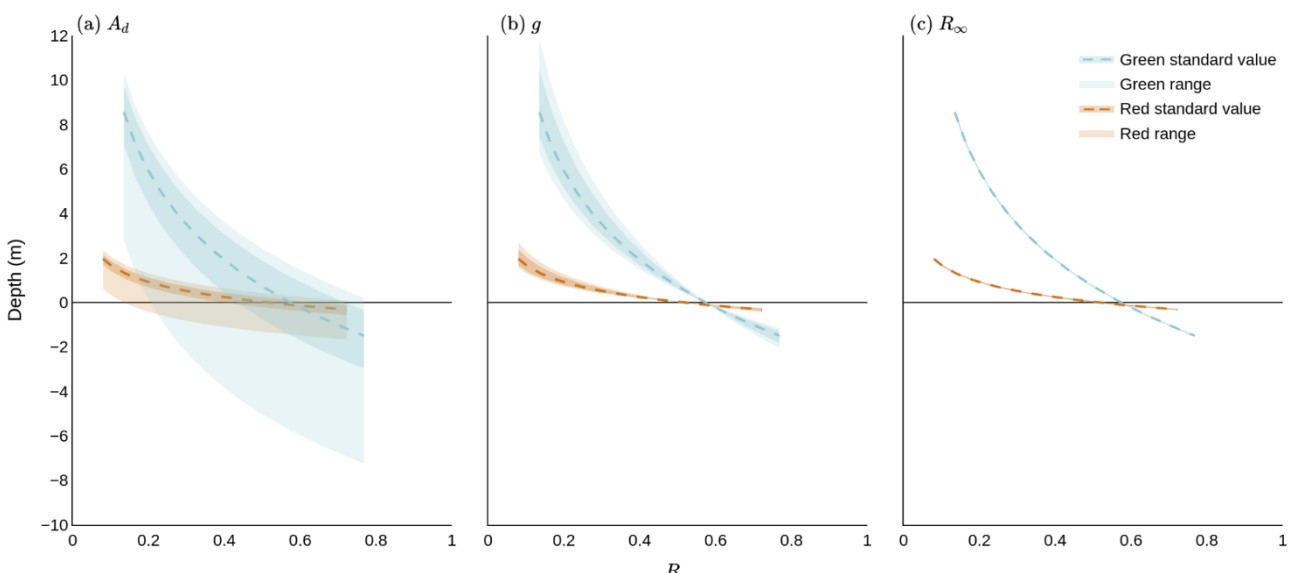

**Figure 6: Sensitivity analysis of the RTE to plausible ranges in its parameter values. Each panel shows the variability of lake depth, as given by Eq. (1), with measured surface reflectance, (a) when $A_d$ is altered only, (b) when $g$ is altered only, and (c) when $R_\infty$ is altered only. When one parameter is altered, the other tuneable parameters are set to their literature value (i.e. $A_d$ is calculated from a ring of pixels, $g = 2K_d$, and $R_\infty$ is the reflectance of the darkest pixel in a neighbouring scene). The range of $R_w$ is defined by the range of water reflectance values that were recorded in the July 8th 2019 inventory (E Glen 2022, personal communication, 22 July). 'Red standard value' and 'Green standard value' are calculated using our tuneable parameter values (i.e. $A_d$ is calculated from a ring of pixels, $g = 2.75K_d$, and $R_\infty$ is the average reflectance of the 10 darkest pixels in a spatiotemporally contiguous scene). 'Red range' and 'Green range' correspond to the possible depths achieved when extreme values (Appendix A.3) are used for the parameter being varied. The uncertainties indicate one standard deviation of the depths achieved by every permutation of the tuneable parameter ranges (Appendix A.3). In (c), the range and uncertainty of the depths are small and appear as thin lines along the standard value lines.**

To investigate the sensitivity of the RTE to the tuneable parameters in the equation, we computed the relationship between depth and $R_w$ across the range of $R_w$ values recorded over supraglacial lakes in our imagery (Fig. 6). We establish that, for the $R_w$ values we observe in the 2019 lake inventory (E Glen 2022, personal communication, 22 July), the red band RTE depths have a theoretical range of -1.66 to 2.68 m whereas the green band RTE depths have a much larger range of -7.23 to 11.76 m. This variation in range helps to explain why the red band RTE plateaus. The maximum depth that could be achieved by the red band RTE on July 8th 2019 from our study region was 2.68 m. The maximum depth retrieved by the red band RTE for any of the five lakes was 2.87 m (Lake 1), which is due to the difference in date and thus the glaciological conditions of the ice sheet. Figure 6 is an indication of the limits to the green and red band RTEs on a specific date, not the absolute limits of the RTE. However, this analysis demonstrates that an empirical limit exists on the depths achievable using both RTE approaches, which is shallower for the red band than it is for the green. The negative depths achieved by both the red and green band RTEs




represent a flaw of the RTE which occurs when the value of $R_w$ is larger than the value of $A_d$. In this case, the calculated lake

bottom albedo is lower than the reflectance of the pixel of interest, leading to a negative depth.

The distribution of green band RTE depths is broader than the distribution of depths for the red band RTE in Fig. 5. We mainly

attribute this to the variation in the range of possible depth values given in Fig. 6a where the green band RTE depth range is

larger than that for the red band RTE. The green band RTE range is larger because of the $A_d$ ranges of the lakes in the 2019

inventory where $A_d$ takes the range of 0.13–0.77 for red and 0.21–0.80 for green. When combined with the difference in band-

dependent values of $R_w$ and $g$, this results in a larger depth range.

## 4 Discussion

The RTE is the most common method applied at scale over the Greenland and Antarctic ice sheets to determine supraglacial

lake depth (Sneed and Hamilton, 2007; Georgiou et al., 2009; Sneed and Hamilton, 2011; Banwell et al., 2014; Pope et al.,

2016; Moussavi et al., 2016; Williamson et al., 2018; Moussavi et al., 2020). The RTE is widely used because the high volume

of optical satellite imagery in the polar regions means locations that are lacking in other types of remotely sensed data can be

studied. However, our analysis has shown that use of the RTE, in its current form, has some limitations. Due to the rapid

attenuation of red light in water, the red band RTE cannot sense deeper than approximately 3 m, with the precise saturation

depths of the lakes being dictated by the $A_d$ value. Therefore, evaluating the RTE with information from the red band means

that depths from the portions of the lakes which are deeper than the red saturation point are being underestimated. As a result,

when the red band RTE is used to calculate lake depth, the total volume of water stored in lakes at the regional ice sheet scale

is also underestimated.

Contrastingly, the use of the green band to evaluate the RTE leads to an overestimation of depth in the deepest portions of the

lakes compared to ICESat-2 and ArcticDEM. The saturation depths of the green band RTE are not visible in Fig. 5 because

the lakes are not deep enough for the physically constrained range of the green band RTE to plateau. However, there are two

distinctive patches in the green band RTE cloud of Fig. 5. These patches are portions of Lake 3 and Lake 5 – which exhibit

some noise within the DEM. Other spatiotemporally contiguous ArcticDEM data are unavailable for these lakes due to the

sampling frequency of the dataset.

The average overestimations of the green band RTE depths are typically larger than the average underestimations of the red

band RTE depths (Table 2), initially lending support to the use of the red band RTE as opposed to the green band RTE.

However, the large variances in volume estimation between the green and red RTE depths and the ArcticDEM DEMs have

contrasting implications. Use of the green band RTE can lead to overestimations of more than 150 % of the lake volume

compared to ArcticDEM DEMs which would lead to subsequent overestimations of surface water volumes on ice sheets if the

green band RTE were to be used at scale. This has further implications for our understanding of the role of meltwater in ice

sheet dynamics and introduces a potential for exaggeration of the contribution of meltwater to localised ice velocities, which

impacts our ability to predict ice calving rates at marine-terminating glaciers (Melton et al., 2022). Contrastingly, use of the



red band RTE can lead to underestimations of 63 % of the lake volume compared to ArcticDEM DEMs. This has the opposite implication for our ability to predict ice calving rates at marine-terminating glaciers, where we could underestimate the contribution of meltwater to localised ice velocities, potentially understating the role of meltwater in ice calving caused by glacier velocity increases.

Due to the plateauing effect of the red band RTE, and the systematic overestimation of the green band RTE, neither parameter selection (ours, nor the literature's) results in good agreement with either ArcticDEM or ICESat-2 for deep lakes. Although previous studies have employed a band-averaging method (Pope et al., 2016; Williamson et al., 2018), we caution against the averaging of the red band RTE depths and the green band RTE depths due to the plateauing effect observed in the red band RTE.

It is inconclusive which parameter selection is best for shallow lakes, highlighting the importance of parameter selection in the use of the RTE. With our parameter value choices, the green band appears to predict lake bathymetry in closer agreement with ArcticDEM and ICESat-2 than the red band. Conversely, using the methods reported in previous literature (Sneed and Hamilton, 2007; Banwell et al., 2014) to calculate each of the parameters leads to the conclusion that the red band RTE, at depths lower than its saturation depth, is more accurate in gauging lake depth than the green band RTE. We suggest that the

reason for this change in the best-performing band, as a result of the different parameter values chosen, is primarily due to changes in the $g$ value as it is clear from Fig. 6 that depth is largely insensitive to the choice of $R_\infty$, and our calculated value of $A_d$ is the same as the value commonly used within existing literature (Sneed and Hamilton, 2007; Moussavi et al., 2020). Specifically, a low coefficient of $K_d$ leads to a low $g$ value which, as found in recent literature (e.g. Pope et al., 2016; Williamson et al., 2017), leads to larger lake depths. When this is combined with the green band in the RTE, it leads to a

significant overestimation of depth which can exceed 5 m compared to the ICESat-2 and ArcticDEM depths (Fig. 2).

Using the parameter values commonly found in the literature (Sneed and Hamilton, 2007; Banwell et al., 2014), the red band RTE predicts depth relatively accurately until it plateaus. This attribute makes it well suited for use with shallower lakes, such as those found on Antarctic ice shelves (Banwell et al., 2014). However, the calculation of $A_d$ needs to be carefully considered. Specifically, if $A_d$ is estimated from a ring of pixels around the edge of the lake (e.g. Sneed and Hamilton, 2007; Moussavi et

al., 2020), then the presence of slush may adversely impact the derivation of a representative value. The differentiation of blue ice from slush on the Antarctic ice sheet is particularly difficult due to their structural and spectral similarities in satellite imagery (Dell et al., 2021). This makes the derivation of $A_d$ even more complicated and care should be taken when calculating $A_d$ in the presence of blue ice due to the potential misidentification of slush. Figure 6 elucidates the importance of the choice of $A_d$ within the RTE, wherein a small change in $A_d$ translates into a large difference in estimated depth.

Although generally in closer agreement with one another than with the red or green band RTEs, ICESat-2 and ArcticDEM cannot be used to track surface water volumes at scale across the ice sheet because of limitations in their spatial and temporal sampling. ICESat-2 acquires elevation measurements along satellite tracks, meaning it cannot be used to measure the entirety of a supraglacial lake. ArcticDEM acquisitions, despite their 2-D nature, are temporally and spatially sporadic, and therefore cannot provide sampling at the frequency required to study lake dynamics. Hence, the RTE is the only method that can feasibly





be applied at scale to monitor the total volume of water held within lakes on the ice sheet surface. However, the consistency between the depths retrieved by ArcticDEM and ICESat-2 could provide value in their use as constraints to the RTE. With a larger amount of ArcticDEM and/or ICESat-2 data, future research could use machine learning to generate a well-constrained depth-detection product from a data-driven, as opposed to a model-derived, approach.

The relatively weaker correlation between the RTE datasets and the observational datasets of ArcticDEM and ICESat-2 is
likely a result of the uncertainty introduced by each of the RTE's tuneable parameters (Fig. 6). $A_d$, in particular, is affected by the potentially incorrect assumption that suspended or particulate matter in the lake is minimal (Sneed and Hamilton, 2011). This raises the issue of cryoconite holes on the ice sheet surface which are known to lower the albedo (Hotaling et al., 2021). Cryoconite holes are formed when aeolian dust settles on the ice sheet. The albedo of the dust-covered area is lower than the surrounding ice, so it heats up and melts the underlying ice, forming vertical shafts. The ponding of surface water partially
cleans these cryoconite holes, resulting in the disbursement of the particulate matter into the lake. However, the method currently used to estimate $A_d$ is assumed to accommodate this potential lowering of lake albedo. Therefore, if cryoconite was present in the lake basin, the ring of pixels used to estimate the lake bottom albedo would likely also contain cryoconite holes. If the water column is affected by particulate matter, this would also affect the value of $g$ (Brodský et al., 2022). Currently, $g$ is calculated from $K_d$, the coefficient for the scattering of downwelling light in the water column. The $K_d$ value is laboratory-
derived from optically clear water, i.e. water that does not contain particulate matter. Consequently, the value of $g$ would be incorrect for lakes which contain particulate matter, further limiting the generalisability of the RTE when it is used in such a scenario.

Currently, the limitations associated with red and green band RTE calculations have wider implications for other areas of cryospheric research, such as calculating hydrofracture likelihood and understanding fluctuations in local ice velocities. Lake
volume is not the only control on the probability of lake hydrofracture, though it is reasonable to assume that the two things should be correlated. However, observational evidence of this correlation remains elusive, despite large-scale study into the phenomena (Williamson et al., 2018). It is possible that these large-scale studies found no evidence of a correlation between hydrofracture and lake volume, at least in part, because of uncertainties in the RTE approach used to derive lake depth.

## 5 Conclusions

The Greenland ice sheet accommodates thousands of supraglacial lakes which form and reform every melt season. Current methods to estimate the volume of these lakes have either relatively poor spatio-temporal sampling or limitations in the accuracy with which they can retrieve lake depth. This study gives a detailed intercomparison of three methods which can be used to estimate lake depth – an integral component in the calculation of lake volume. Tracking the volume of water storage on the surface of the ice sheet is important for quantifying hydrofracture likelihood and determining the impacts of lake
drainage on ice sheet velocities, and requires ice sheet-wide coverage and high temporal sampling to resolve seasonal dynamics.



Within this study, we found that two of the three methods considered, namely the ArcticDEM DEMs and the ICESat-2 laser altimetry approaches, have close agreement. However, these methods are spatially and temporally restricted, meaning they cannot be used to derive comprehensive estimates of surface water storage at the ice sheet scale. Our third method, which uses the Philpot (1987) RTE to derive depth from optical imagery, has relatively poor agreement with the other two methods, especially for deeper supraglacial lakes. We detected a plateauing effect in the red band RTE which is caused by the rapid attenuation of light in the red band, suggesting the use of this method will consistently underestimate the depths of lakes which are deeper than the lake-specific saturation limit. Within this study, the use of our RTE parameter values improved the ability of the green band RTE to sense lake depth, and a comprehensive sensitivity analysis of the RTE's tuneable parameters leads us to believe that further alterations to the parameter calculation and/or equation could be undertaken to improve the method. Interestingly, the methods currently used within the literature to determine the parameter values appear to hamper the accurate calculation of lake depth using the green band RTE. However, this is a case study of five lakes on the southwest Greenland ice sheet, and further work is required to understand whether this conclusion is generalisable to the whole ice sheet.

Nonetheless, the RTE is the only method which can currently be deployed at an ice sheet scale due to data availability constraints, meaning improvements to the method are paramount to its potential use as an accurate method for calculating lake depth. We suggest that future improvements to the current equation should focus on the calculation of $A_d$ which has been shown to have the greatest influence on the derived depths. However, the calculation of $A_d$ also poses significant technical difficulties due to the issues in differentiating between water and ice in satellite imagery, so care must be taken to ensure that new methods are robust and replicable at the ice sheet scale.

**Appendix A**

**Section 1: Characteristics to reduce the 2019 lake inventory**

A 2019 inventory (E Glen 2022, personal communication, 22 July) of the maximum areal extents of all water bodies in the study region was used as the basis for selecting our five lakes, with the following characteristics used as the selection criteria:

- The water body is intersected by an ICESat-2 reference ground track (removed 7,519 water bodies);
- The seasonal maximum water body area is greater than 1 km$^2$ but less than 10 km$^2$. This removes small water bodies which are absent in low melt years, and large water bodies which are formed by the merging of smaller water bodies, thus leaving supraglacial lakes with dimensions that were representative of the regional average (removed a further 338 water bodies); and,
- The water body circumference is less than 30 km i.e. it is not a highly elongated feature such as a stream (removed a further 28 water bodies).

These characteristics reduced the 2019 inventory from 7,913 to 28. The lakes were then considered for their ICESat-2 data quality, where the highest quality translates to the basins which can be most easily delineated from ICESat-2 photon refraction (Sect. 2.3). Additionally, the 28 lakes were visually appraised for their level of activity (Sect. 2.2.1) to ensure they drained and



refilled. These comparisons identified five lakes for which depth could be derived for all three measurement techniques. The
other lake basins from the initial subset of 28 could not easily be identified by ICESat-2 or could not be resolved using the
ArcticDEM digital elevation models (Sect. 2.2.1).

**Section 2: Input data**

**Table A1: ArcticDEM, ICESat-2 and Sentinel-2 data used for depth retrieval in each of the five lakes. Acquisition dates are in bold text and are in the format YYYYMMDD.**

| Lake number | Dataset | Filename |
|---|---|---|
| 1 | ArcticDEM tile | SETSM_s2s041_WV01_**20200304**_1020010093157200_1020010093C80200_2m_lsf_seg1_dem |
| | ICESat-2 track | ATL03_**20200706**005932_01630805_003_01 |
| | Sentinel-2 tile | T22WEV_**20200702**T150759 |
| 2 | ArcticDEM tile | SETSM_s2s041_WV02_**20210312**_10300100BB24B100_10300100BBC0A100_2m_lsf_seg1_dem |
| | ICESat-2 track | ATL03_**20200717**114945_03380803_003_01 |
| | Sentinel-2 tile | T22WEV_**20200717**T150921 |
| 3 | ArcticDEM tile | SETSM_s2s041_WV01_**20200304**_1020010093157200_1020010093C80200_2m_lsf_seg1_dem |
| | ICESat-2 track | ATL03_**20200706**005932_01630805_003_01 |
| | Sentinel-2 tile | T22WEV_**20200704**T145921 |
| 4 | ArcticDEM tile | SETSM_s2s041_WV01_**20200511**_1020010094C9D900_1020010098791800_2m_lsf_seg3_dem |





| | | |
|---|---|---|
| ICESat-2 track | ATL03_**20200706**005932_01630805_003_01 | |
| Sentinel-2 tile | T22WEA_**20200704**T145921 | |
| | ArcticDEM tile | SETSM_s2s041_WV01_**20200620**_10200100982C5E00_102001009793BD00_2m_lsf_seg1_dem |
| 5 | ICESat-2 track | ATL03_**20190716**051841_02770403_003_01 |
| | Sentinel-2 tile | T22WFV_**20190725**T150015 |


**Section 3: Selection of the RTE tuneable parameter values**

Commonly, within the radiative transfer equation (RTE) the derivation of $A_d$ is either specific to a lake, or a regional average is used (Sneed and Hamilton, 2007). We have used an individual $A_d$ for each lake because the reflectance of pixels surrounding supraglacial lakes can vary considerably. In our study, we calculated the $A_d$ values specific to each lake. The red $A_d$ values

were 0.42, 0.46, 0.35, 0.58, and 0.57 for lakes 1, 2, 3, 4, and 5 respectively and the green $A_d$ values were 0.45, 0.49, 0.40, 0.60, and 0.58 for lakes 1, 2, 3, 4, and 5 respectively.

Our derivation of $g$ is given by 2.75 $K_d$. In addition to this being an average of the potential range of $g$ (1.5 $K_d$ – 3 $K_d$), this coefficient of $K_d$ incorporates the concept that the value of $K_d$ is dependent on depth (Kirk, 1989). This means an average value can be expected to retrieve a more representative depth than either the lowest or highest values in the $K_d$ range.

Typically, $R_\infty$ is calculated as the reflectance of the darkest pixel in a scene containing optically deep water (Sneed and Hamilton, 2007). Optically deep water, in the case of the Greenland ice sheet, consists of open ocean pixels. To reduce the impact of atmospheric effects, $R_\infty$ is ideally calculated from either the same scene as the one containing the pixels of interest where there are open ocean pixels, or a concurrent neighbouring scene. However, it is not always possible to sample $R_\infty$ from a concurrent neighbouring scene due to the location of the pixels of interest, and/or cloud cover. In this case, a non-concurrent

and/or non-neighbouring scene is chosen instead (Table A2). Sneed and Hamilton (2011) argued that all optically deep water has similar spectral characteristics and therefore the precise method for determining this value negligibly affects the depths derived using the RTE. The findings of our RTE tuneable parameter sensitivity analysis agree with Sneed and Hamilton (2011) (Fig. 6).

**Table A2: Sentinel-2 tiles used to retrieve $R_\infty$ values for each lake. Acquisition dates are in bold text and are in the format YYYYMMDD.**





| Lake number | Sentinel-2 tile |
|---|---|
| 1 | T21WXP_**20200702**T150759 |
| 2 | T21WXP_**20200717**T150921 |
| 3 | T20WPT_**20200704**T154819 |
| 4 | T20WPT_**20200704**T154819 |
| 5 | T22VDQ_**20190712**T144759 |
| July 8th 2019 lake inventory | T21WXN_**20190708**T150809 |

**Section 4: Plausible ranges of the RTE tuneable parameters**

To calculate the uncertainty of the red and green band RTEs for the study lakes, we had to first understand the plausible ranges

of the three tuneable parameters. The range of $A_d$ was calculated as the range of $R_w$ values of every pixel in the 30 m ring of pixels around each lake, as detailed in Sect. 2.2.1. The range of $g$ was 1.5 to 3 $K_d$ at 0.1 intervals for every lake, where $K_d$ was calculated as the band-specific solution of Eq. (3) for the average $a_w$ and $b_m^{fw}$ values from Smith and Baker (1981) for both the red and green optical bands of Sentinel-2. We calculated the ranges of $R_\infty$ from the Sentinel-2 tiles detailed in Table A1. We manually appraised these scenes for spurious values caused by aeroplane overpasses and sediment contamination with the aid

of a band combination to highlight snow and clouds (Band 1, Band 11, and Band 12 (Coastal and aerosol, shortwave infrared (1610 nm), and shortwave infrared 2190 nm)). The $R_\infty$ ranges consist of the $R_w$ values of the ten darkest pixels from each scene which were true dark sea pixels.

Our study includes data from the sensitivity analysis that we carried out on the tuneable parameters using the plausible ranges

of $R_w$ from the July 8th 2019 lake inventory (E Glen 2022, personal communication, 22 July) (Fig. 6) (Table A3). The method of calculating $A_d$ was slightly different due to the scale of the data. We calculated the range of $A_d$ as the range of the average $R_w$ values of the 30 m rings of pixels around all of the lakes in the 2019 lake inventory. We used the same method to calculate $g$ and $R_\infty$ as that which we used to calculate the uncertainty for the five study lakes. The scene from which we calculated the $R_\infty$ range was spatiotemporally contiguous with the lake inventory data (Table A2).


**Table A3: The ranges of the tuneable parameters used to find the uncertainty of the 2019 lake inventory RTE depths**



| Parameter | Band | Value range |
|---|---|---|
| $A_d$ | Red | 0.1347–0.7724 |
| | Green | 0.2055–0.7973 |
| $g$ | Red | $XK_d$ where X = 2, 2.1, 2.2, 2.3, 2.4, 2.5, 2.6, 2.7, 2.8, 2.9, 3, 3.1, 3.2, 3.3, 3.4, and 3.5, and $K_d$ = 0.4075875 |
| | Green | $XK_d$ where X = 2, 2.1, 2.2, 2.3, 2.4, 2.5, 2.6, 2.7, 2.8, 2.9, 3, 3.1, 3.2, 3.3, 3.4, and 3.5, and $K_d$ = 0.07636 |
| $R_\infty$ | Red | 0.0254–0.0260 (0.0254, 0.0257, 0.0258, 0.0258, 0.0258, 0.0259, 0.0259, 0.0260, 0.0260, 0.0260) |
| | Green | 0.0474–0.0479 (0.0474, 0.0475, 0.0477, 0.0478, 0.0478, 0.0478, 0.0479, 0.0479, 0.0479, 0.0479) |

## Author contributions

LM and AL conceptualised the research. AL, MM, and JM provided LM with supervision. LM, AL, MM, and JM designed the study. JB wrote and supplied the DEM stacking script detailed in Sect. 2.3. EG provided the 2019 lake inventory shapefiles. MW, LSS and RLA processed the ICESat-2 ATL03 data. LM obtained the data, performed the analyses, created all figures and wrote the manuscript. All co-authors contributed to manuscript editing.

## Competing interests

One author is a member of the editorial board of The Cryosphere. The peer-review process was guided by an independent editor, and the authors have no other competing interests to declare.

## Acknowledgements

This study was supported by the POLAR+ 4DGreenland project, which was funded by the European Space Agency (ESA) via ESA Contract No. 4000132139/20/I-EF. Additional support came from MII Greenland via grant No. NE/S011390/1. MM was supported by the UK NERC Centre for Polar Observation and Modelling, and the Lancaster University-UKCEH Centre of Excellence in Environmental Data Science.



We would like to extend our thanks to the UK NERC Centre for Polar Observation and Modelling (CPOM), and the whole POLAR+ 4DGreenland team for their support.

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
