# Peer review of "Evaluation of satellite methods for estimating supraglacial lake depth in southwest Greenland"

_The Cryosphere, 2023_

## Referee Comment (RC2)

**Review of tc-2023-103**
**Evaluation of satellite methods for estimating supraglacial lake depth in southwest Greenland**

**Summary**
This paper offers methodological insight into the effectiveness of the radiative transfer equation (RTE) for calculating supraglacial lake depths. It does this by comparing the RTE using the red band and green band from Sentinel-2 against lake depths calculated from both ICESat-2 and ArcticDEM. Overall, I think this is an excellent paper, which is well written and contributes significantly to the field of remote sensing in glaciology. However, there are some major and minor comments that I would like to be addressed prior to publication.

**Major Comments**

The methods section for the RTE method is very difficult to follow. I think it would help to start by explaining that until now, there have been values for various parameters that have been used in the literature, and to state these first. You can then go on to justify why you look to create and use new values.

The above comment cascades into Figure 6 and the caption for Figure 6 being very hard to follow. Please think about how you can better explain and present the variables tried and tested. It may be that some information from Appendix Section 4 would be better placed in the main text.  Following on from this, whilst in Figure 6 you vary each parameter in turn, what happens if you vary all parameters together, is this not something that needs testing?

More attention should have been paid to the selection of values of Ad in the RTE. Specifically, I would recommend the authors test a wider range of Ad values before this work is published. The author should read and refer to Dell et al. (2020), who use careful selection of Ad values to avoid the impact of slush.

Despite your comments r.e. averaging the red and the green band depths, I would like to see some evidence for this method not being suitable moving forward, particularly given its use in both Pope et al. (2016) and Williamson et al. (2018). I understand what you are saying with regards to the plateauing effect caused using the red band, but it would be more convincing if you could provide evidence for this. You also do not mention the fact that previous studies have averaged these depths until the discussion, I would advise mentioning this much earlier on.

**Minor Comments**

L74: Does this region only contain active lakes?

L134-136: For the section starting 'we manually appraised', please can you better clarify what was done here, I can't make sense of it.

L160: Whilst I understand that it is more likely for lake bathymetry to change over 11 months for Lake 5, surely the lake bathymetry could feasibly change over any lake and time period?

L165: For the paragraph containing line 165 (which details the method used to calculate each lake's depth from the DEM) and the two following paragraphs, much more detail is needed. It is very unclear how you carried out your methodology.

L194: lake surface?

L269-L275: You only talk about the red depths in detail here, what about the green depths?

L325: I am not sure 'However' is the right word to use here.

L332-224: This section could do with some re-wording to improve its clarity.

L357: The authors should also consider referencing Moussavi et al. (2020) here.

**Figures**

Figure 1: Caption – where are the background images from? Sentinel-2?
North Arrow is hard to see in main map and subset maps.

Figure 3: For the Y-axis find a way to space out the text for 'ArcticDEM' and 'ICESat-2'

Figure 4: Please add North labels and scale bars to these plots! You also need to state that you show the ICESat-2 transects on the depth plots.

**Appendices**

Appendix A Section 1: The authors should consider moving comments on the quality of ICESat-2 into the main text.

Appendix A Section 2:
Table A1: What date was the imagery downloaded by you? – Apply this comment elsewhere too.

**References in this review**

Dell, R., Arnold, N., Willis, I., Banwell, A., Williamson, A., Pritchard, H., & Orr, A. (2020). Lateral meltwater transfer across an Antarctic ice shelf. *The Cryosphere*, *14*(7), 2313-2330.

Moussavi, M., Pope, A., Halberstadt, A. R. W., Trusel, L. D., Cioffi, L., & Abdalati, W. (2020). Antarctic supraglacial lake detection using Landsat 8 and Sentinel-2 imagery: Towards continental generation of lake volumes. *Remote Sensing*, *12*(1), 134.

Pope, A., Scambos, T. A., Moussavi, M., Tedesco, M., Willis, M., Shean, D., & Grigsby, S. (2016). Estimating supraglacial lake depth in West Greenland using Landsat 8 and comparison with other multispectral methods. *The Cryosphere, 10*(1), 15-27.

Williamson, A. G., Banwell, A. F., Willis, I. C., & Arnold, N. S. (2018). Dual-satellite (Sentinel-2 and Landsat 8) remote sensing of supraglacial lakes in Greenland. *The Cryosphere*, *12*(9), 3045-3065.

---

## Author Comment (AC1)

Point-by-point reply to Reviewer 1 – Manuscript entitled: Evaluation of satellite methods for estimating supraglacial lake depth in southwest Greenland, authored: Melling et al.

Manuscript iteration: Revision post review

Editor: Joseph MacGregor

We thank anonymous reviewer #1 for their comments on and suggested edits to our paper. These are listed below in bold type, with our response to each one given in normal type. We note that all line numbers refer to the original manuscript.

**1) "First, the conclusions suggest using ICESat-2 to constrain or correct the RTE methods. Although they do not use machine learning, Datta et al., (2021) uses ICESat-2 lake depths to constrain empirically-derived depths from Landsat-8, among other imagery sources. They also do not use the RTE equation used here, but I still think it is worth elaborating on how this study builds upon theirs (or how results from both could benefit the community)."**

We agree that Datta and Wouters (2021) is worth elaborating on and propose to add/change lines 364-368 to read, *"Methods which exploit regularly acquired 2-D satellite imagery - such as the application of the RTE to optical satellite imagery - are thus needed to monitor the total volume of water held within lakes on the ice sheet surface and its evolution through time. ArcticDEM and ICESat-2 data are of most value for their potential to constrain these methods. For example, Datta and Wouters (2021) used ICESat-2 to constrain empirically derived estimates of lake bathymetry from Sentinel-2 scenes in western Greenland. With a larger amount of ArcticDEM and/or ICESat-2 data, we suggest that future research could combine multiple satellite bands (Adegun et al., 2023) and data sources as inputs to a machine learning model and generate a well-constrained depth-detection product using a data-driven approach, as opposed to the model-derived approach we use here."*

We will also add the Adegun et al. (2023) citation to the reference list of the revised manuscript.

**2) "On the subject of ICESat-2, its value for this study is not immediately clear to me. I assume that it serves as high-accuracy validation for the RTE and ArcticDEM methods, but this is not mentioned explicitly in the text. Also, ICESat-2 is barely discussed after Section 3.1. Between these issues and the sparse coverage over all five lakes (especially Lakes 4/5), the manuscript needs to provide more justification on why the ICESat-2 depths are useful here."**

Due to its spatially limited nature, ICESat-2 can only be used in a comparison along 1-D transects. As such, we do not refer to it in sections 3.2 and 3.3 as these sections intercompare the RTE and ArcticDEM in 2-D. The main utility of using three different depth detection methods rather than two is that it increases our confidence in depths calculated at locations where there is agreement between methods. Where two methods disagree, each one is equally likely to be 'correct'. Bringing in a third method in this case helps to identify in which of the first two methods possible deficiencies lie.

We use it here in addition to ArcticDEM because it is considered to have a high level of accuracy (e.g. Markus et al., 2017).

At line 67, we will add "*Intercomparing multiple depth detection methods increases our confidence in the depths calculated at locations where there is agreement between methods. This is especially important in the absence of 'ground truth' data*."

**3) "Page 2, Line 32: For a general audience, I recommend mentioning why lakes prefer draining over refreezing (or vice versa)."**

In the revised manuscript, we propose to add the following text in lines 31-33, *"These lakes either drain or refreeze, with ~34 % of lakes at lower elevations draining slowly, ~14 % draining rapidly and ~50 % refreezing. At higher elevations, lakes tend to refreeze (Selmes et al., 2013)."*

**4) "Page 2, Lines 33-36: Hydrofracturing is formally defined on Line 36, so I would refrain from mentioning it until then."**

We propose editing lines 33-36 to read, "*This process is known as hydrofracture, and related drainage events can occur in as little as two hours (Das et al., 2008). In these events, the water is routed to the base of the ice sheet where it can cause a hydraulic pressure increase that temporarily lifts the ice off the bed. This process can enhance basal sliding and increase ice flow rates*"

**5) "Page 2, Line 45: I would like to see a reference or two here demonstrating the importance of melt lakes for models, if possible."**

We will add references to Christoffersen et al. (2018) (https://doi.org/10.1038/s41467-018-03420-8) and Tedesco et al. (2013) (https://doi.org/10.1088/1748-9326/8/3/034007) on line 45.

**6) "Page 2, Line 51: Replace semi-colon with colon."**

We will replace the semi-colon on line 51 with a colon.

**7) " Figure 1: Nice figure! For panels 1-5, I would specify in the caption where the imagery is from."**

In the caption for Figure 1, we will add the text, "*background images in panels (1)-(5) are the Sentinel-2 tiles detailed in Table A1.*"

**8) "Page 4, Lines 92-93: This sentence doesn't add much justification – the following sentence is enough."**

We will edit lines 92-95 to read, "*These bands have previously been used to determine lake depth on the Greenland ice sheet (e.g. Williamson et al., 2018; Moussavi et al., 2020; Datta and Wouters, 2021)*."

**9) "Page 6, first paragraph: This is overall well-written, but it is also getting a bit in the weeds. I suggest condensing it to something like: "Previous studies assumed that Ku≈ 1-2.5Kd, with Brodsky**

**et al., (2022) suggesting that higher Ku values (and therefore higher g values) lead to more accurate lake depths. Here, we use an average of the above range and take Ku = 1.75Kd, or g = 2.75Kd.""**

We agree that this reads better when condensed and propose editing lines 124-129 to read, "*Many laboratory-derived estimates exist of $K_d$ but very few exist of $K_u$ (Philpot, 1989). Other studies have taken $K_u$ to be equal to $K_d$, and thus g to be $2K_d$ (e.g. Maritorena et al., 1994; Sneed and Hamilton, 2007), but $K_u$ must be larger than $K_d$ because upwelling photons are more rapidly attenuated than the downwelling photon flux in water (Kirk, 1989). Experimental studies suggest that g could be as high as $3.5K_d$ (Kirk, 1989), with some studies suggesting a higher g value leads to more accurate lake depths (Brodský et al., 2022). Here, we therefore use an average of this range and take $g = 2.75K_d$.*"

**10) "Page 7, Line 183: Outdated reference. Refer instead to Markus et al., (2017) – see "New References" section below for full citation."**

We will refer instead to Markus et al. (2017) on line 183. We will also remove the reference to Abdalati et al. (2010) from the reference list and add the citation for Markus et al. (2017).

**11) "Section 2.4, first paragraph: I suggest noting that the spacing between ICESat-2 beam pairs is 3.3 km, which limits the coverage of individual lakes."**

We propose the addition of "*The spacing between ICESat-2 beam pairs at all latitudes is ~3.3 km which limits the coverage of individual lakes.*" at line 185.

**12) "Page 7, Line 186: Cite Neumann et al., (2019) with the mention of ATL03 usage."**

In altering line 186 to add a citation of Neumann et al. (2019), we noticed that the version number was not immediately clear to a reader unfamiliar with ICESat-2 so propose to alter the line to read, "*We estimate the lake bathymetry of the supraglacial lakes using the ICESat-2 ATLAS ATL03 (version 3) data product (Table A1) (Neumann et al., 2019)…*". We will also add the Neumann et al. (2019) citation to the reference list.

**13) "Page 9, Lines 208-209: Not sure if I follow the logic here. Why not keep ICESat-2 at its native resolution, if the other datasets are resampled to 0.7 m?"**

The ICESat-2 bathymetry data set is digitised manually by 10 experts based on the ICESat-2 photon cloud, and subsequently averaged to minimise observer error, and so does not have a native resolution as such. Individual experts sampled with different frequencies along the photon cloud and a final sampling of 100 equidistant points per lake was selected to accommodate differences in manual delineations. Since the obtained bathymetry is smooth, using another sampling interval would not make a big difference to Figure 2 or any results obtained from the subsequent analysis of its data.

To clarify the manual delineation procedure, we will alter lines 192-196 to read, "*We invited 10 altimetry experts to manually digitise the lake bathymetry from the refraction-corrected ATLAS ATL03 photon data plots using an online digitisation tool (https://apps.automeris.io/wpd/).*" There will be no effect on Figure 3 as the data for this figure were extracted from each dataset at the 100 equidistant ICESat-2 points along each transect. To reflect this, we will add the following text at line 226,

*"…coefficient for each method pairing at each of the 100 equally spaced points along which the ICESat-2 data was sampled (Fig. 3)."*

**14) "Table 1: Given Figure 3, I don't think this table adds much to the paper. I think a table showcasing the maximum depth for each lake and method would be more useful."**

We agree with the reviewer that this table does not currently add much to the paper. In the revised manuscript, we will alter this table to display the maximum depths for each method for each lake, and a column showing which method has achieved the maximum depth for each lake.

**15) "Figure 4: This is a really nice figure, and I think it merits more discussion. In particular, I notice DEM/RTE differences that seem to be related to depth. Also, the green band has expectedly large underestimations in a few spots for Lakes 4 and 5. I would like to see some speculation in the Results or Discussion on these points."**

In the revised manuscript, we propose to add the following text at line 254, *"Consistent with the findings from Fig. 2, the lakes exhibit a relationship between the green band RTE depths and the ArcticDEM depths where the green band RTE overestimates depth in the deepest portions of the lakes. This is particularly evident in Lake 5 as its bathymetry is simpler than that of the other lakes. Additionally, there are notable depth underestimations of the green band RTE in Lake 4 and Lake 5. These underestimations correspond to floating ice on the lake surface which is not present in the ArcticDEM data."*

**16) "Page 15, Lines 269-270: For visual reference, I suggest pointing out the "plateau depths" in Figure 5, using a dotted line or marker(s)."**

In the revised manuscript, we will add dashed lines to Figure 5 to point out the plateau depths for visual reference. Additionally, we will edit the caption to read (at line 267), *"The diagonal long-dashed lines represent one-to-one agreements between the depth datasets. The red-band RTE plateau depths are indicated by the labelled short-dash white lines in (a)."*

**17) "Figure 6 caption, first sentence: Suggest rephrasing to "Sensitivity analysis of RTE parameters, with plausible values given for each.""**

We propose altering line 278 to read, *"Figure 6: Sensitivity analysis of the RTE parameters, within plausible ranges identified for each parameter (Appendix A.3)."*

**18) "Lines 298-300: It might be a bit much to call this a flaw in the method. Water reflectance is very low (~0.1 in the red band) unless specular reflection is observed, so you would need a very dirty lake bottom to achieve negative lake depths. This could be a more feasible issue for the green band, but I would imagine that it is still very uncommon."**

We agree that this wording was too strong and will alter lines 298-300 to read, *"Both the red and green band RTEs produce negative depths when the value of $R_w$ is larger than the value of $A_d$. Physically, this*

*means that the lake bottom albedo is lower than the reflectance of the pixel of interest. In practice, this only occurs in scenarios where; a) the pixel of interest represents misclassified floating ice, such as in the green band RTE plots for Lake 4 and Lake 5 (Fig. 4), or b) as a result of uncertainty in $A_d$.*"

**19) "Page 17, Lines 318-320: Just curious, what is a saturation depth for the green band? A ballpark number is sufficient."**

At line 318, we propose to add the following text, "*From Fig. 6 we can see that the saturation depth of the green band RTE is approximately 8-11 m. This depth is dependent upon the values of $A_d$ and thus will be different for every lake.*"

**20) "Page 17, Lines 326-328: This sentence has redundant wording. I suggest revising to something like, "We determined that use of the green band RTE can lead to lake volume overestimations of more than 150% relative to ArcticDEM, with similar overestimations expected at larger scales.""**

We will edit lines 326-328 to read, "*Use of the green band RTE can lead to lake volume overestimations of 150 % relative to ArcticDEM, with similar overestimations expected at larger scales.*" in the revised manuscript.

**21) "Page 18, Line 339: "…and consistent overestimations in the green band.""**

We propose to edit the text on lines 338-339 to read, "*…red band RTE depths and the green band RTE depths due to the plateauing effect observed in the red band RTE and consistent overestimations in the green band RTE.*"

**22) "Page 18, Lines 344-347: Long sentence, needs to be more concise (or split into two sentences)."**

In the revised manuscript, lines 344-347 will read, "*It is clear from Fig. 6 that depth is largely insensitive to the choice of $R_\infty$. Since our calculation of $A_d$ is the same as that which is commonly used within existing literature (Sneed and Hamilton, 2007; Moussavi et al., 2020) we suggest that there is disagreement with respect to the best-performing band, at depths lower than the saturation point of the red band RTE, because we use a different value of g.*"

**23) "Page 20, Lines 406-407: Given that only five lakes were observed, this is a rather strong conclusion to make."**

We agree that this assertion was too strong given the limited number of observed lakes and so propose altering lines 406-407 to read, "*Interestingly, the methods currently used within the literature to determine the parameter values appear to limit the accuracy of lake depth calculation using the green band RTE within our five-lake sample.*"

**24) "Appendix A, Section 1: "Characteristics" ---> "Criteria""**

We will change "*Characteristics*" in the section heading of Appendix A Section 1 to read, "*Criteria*". Additionally, on line 173, we will change "*characteristics*" to "*criteria*".

**25) "Table A1: The ICESat-2 data used for this study is out of date – Version 006 was released in June. As a sanity check, I would see if there's any significant differences in the ICESat-2 V003/V006 data over the five lakes."**

As the reviewer correctly points out, the analysis is based on an older version (version 3) of the ICESat-2 data set. This information will be added to the text (line. 186 will read, "*…using the ICESat-2 ATLAS ATL03 (version 3) data product…*").

We further noticed that information on which ICESat-2 tracks were used for the different lakes was missing from the manuscript and we will therefore add it to Table A1 in the revised manuscript.

We have examined the two datasets over our five lakes and have determined that:

- The geolocation of the satellite track has moved by 0.6 - 1.2 m in version 6 (Fig. 1). This is within the spacing of the ArcticDEM and S2 datasets and so should not affect our comparison.

[Figure]

*Figure 1: The geolocation difference between ICESat-2 version 3 and ICESat-2 version 6.*

- The individual photon heights have changed slightly on the order of 0.2 - 0.5 m (Figs. 2 – 6). This offset is observed both for the lake surface and bed photons so this should not affect lake depth estimates (which subtract the latter from the former), especially considering the uncertainty associated with the manual delineations.

[Figure]

*Figure 2: The photon plots of ICESat-2 version 3 (ATL03 v3) and ICESat-2 version 6 (ATL03 v6) for Lake 1.*

[Figure]

*Figure 3: The photon plots of ICESat-2 version 3 (ATL03 v3) and ICESat-2 version 6 (ATL03 v6) for Lake 2.*

[Figure]

*Figure 4: The photon plots of ICESat-2 version 3 (ATL03 v3) and ICESat-2 version 6 (ATL03 v6) for Lake 3.*

[Figure]

*Figure 5: The photon plots of ICESat-2 version 3 (ATL03 v3) and ICESat-2 version 6 (ATL03 v6) for Lake 4.*

[Figure]

*Figure 6: The photon plots of ICESat-2 version 3 (ATL03 v3) and ICESat-2 version 6 (ATL03 v6) for Lake 5.*

References within this response

Abdalati, W., Zwally, H.J., Bindschadler, R., Csatho, B., Farrell, S.L., Fricker, H.A., Harding, D., Kwok, R., Lefsky, M., Markus, T. and Marshak, A.: The ICESat-2 laser altimetry mission, P. IEEE, 98, 735-751, doi:10.1109/JPROC.2009.2034765, 2010.

Adegun, A.A., Viriri, S. & Tapamo, JR.: Review of deep learning methods for remote sensing satellite images classification: experimental survey and comparative analysis, J. Big Data, 10, 93, doi:10.1186/s40537-023-00772-x, 2023.

Brodský, L., Vilímek, V., Šobr, M. and Kroczek, T.: The Effect of Suspended Particulate Matter on the Supraglacial Lake Depth Retrieval from Optical Data, Remote Sens-Basel, 14, 5988, doi:10.3390/rs14235988, 2022.

Christoffersen, P., Bougamont, M., Hubbard, A., Doyle, S.H., Grigsby, S. and Pettersson, R.: Cascading lake drainage on the Greenland Ice Sheet triggered by tensile shock and fracture. Nat. Commun., 9, 1064, doi:10.1038/s41467-018-03420-8, 2018.

Das, S., Joughin, I., Behn, M., Howat, I., King, M., Lizarralde, D. and Bhatia, M.: Fracture propagation to the base of the Greenland Ice Sheet during supraglacial lake drainage, Science, 320, 778-781, doi:10.1126/science.1153360, 2008.

Datta, R.T. and Wouters, B.: Supraglacial lake bathymetry automatically derived from ICESat-2 constraining lake depth estimates from multi-source satellite imagery, Cryosphere, 15, 5115-5132, doi:10.5194/tc-15-5115-2021, 2021.

Kirk, J.T.O.: The upwelling light stream in natural waters, Limnol. Oceanogr., 34, 1410-1425, doi:10.4319/lo.1989.34.8.1410, 1989.

Maritorena, S., Morel, A. and Gentili, B.: Diffuse reflectance of oceanic shallow waters: Influence of water depth and bottom albedo, Limnol. Oceanogr, 39, 1689-1703, doi:10.4319/lo.1994.39.7.1689, 1994.

Markus, T., Neumann, T., Martino, A., Abdalati, W., Brunt, K., Csatho, B., Farrell, S., Fricker, H., Gardner, A., Harding, D. and Jasinski, M.: The Ice, Cloud, and land Elevation Satellite-2 (ICESat-2): science requirements, concept, and implementation, Remote Sens. Environ, 190, 260-273, doi: 10.1016/j.rse.2016.12.029, 2017.

Moussavi, M., Pope, A., Halberstadt, A., Trusel, L., Cioffi, L. and Abdalati, W.: Antarctic supraglacial lake detection using Landsat 8 and Sentinel-2 imagery: Towards continental generation of lake volumes, Remote Sens-Basel, 12, 134, doi:10.3390/rs12010134, 2020.

Neumann, T., Martino, A., Markus, T., Bae, S., Bock, M., Brenner, A., Brunt, K., Cavanaugh, J., Fernandes, S., Hancock, D. and Harbeck, K.: The Ice, Cloud, and Land Elevation Satellite–2 Mission: A global geolocated photon product derived from the advanced topographic laser altimeter system, Remote Sens. Environ., 233, 111325, doi.org/10.1016/j.rse.2019.111325, 2019.

Philpot, W.: Bathymetric mapping with passive multispectral imagery, Appl. Optics, 28, 1569-1578, doi:10.1364/ao.28.001569, 1989.

Selmes, N., Murray, T. and James, T.D.: Characterizing supraglacial lake drainage and freezing on the Greenland Ice Sheet, Cryosphere Discussions, 7, 475-505, doi:10.5194/tcd-7-475-2013, 2013.

Sneed, W. and Hamilton, G.: Evolution of melt pond volume on the surface of the Greenland Ice Sheet, Geophys. Res. Lett., 34, L03501, doi:10.1029/2006gl028697, 2007.

Tedesco, M., Willis, I.C., Hoffman, M.J., Banwell, A.F., Alexander, P. and Arnold, N.S.: Ice dynamic response to two modes of surface lake drainage on the Greenland ice sheet, Environ. Res. Lett., 8, 034007, doi:10.1088/1748-9326/8/3/034007, 2013.

Williamson, A., Banwell, A., Willis, I. and Arnold, N.: Dual-satellite (Sentinel-2 and Landsat 8) remote sensing of supraglacial lakes in Greenland, Cryosphere, 12, 3045-3065, doi:10.5194/tc-12-3045-2018, 2018.

---

## Author Comment (AC2)

Manuscript iteration: Revision post review

Editor: Joseph MacGregor

We thank anonymous reviewer #2 for their comments on and suggested edits to our paper. Please find each comment given below in bold type, with our response following in normal text. All line numbers refer to the original manuscript.

**1) "The methods section for the RTE method is very difficult to follow. I think it would help to start by explaining that until now, there have been values for various parameters that have been used in the literature, and to state these first. You can then go on to justify why you look to create and use new values."**

We agree with the reviewer that this section was difficult to follow. In the revised manuscript we will reorder the section.

**2) "The above comment cascades into Figure 6 and the caption for Figure 6 being very hard to follow. Please think about how you can better explain and present the variables tried and tested. It may be that some information from Appendix Section 4 would be better placed in the main text. Following on from this, whilst in Figure 6 you vary each parameter in turn, what happens if you vary all parameters together, is this not something that needs testing?"**

In the revised manuscript we will alter the text for the Figure 6 caption to read, "*Figure 6: Sensitivity analysis of the RTE parameters, within plausible ranges identified for each parameter (Appendix A.3). Each panel shows the variability of lake depth, as given by Eq. (1), with measured surface reflectance, (a) when $A_d$ is altered only, (b) when g is altered only, and (c) when $R_\infty$ is altered only. When one parameter is altered, the other tuneable parameters are set to their 'literature value' (see Method section). $R_w$ is varied over its observed range on July 8th 2019, where reflectance values were extracted using a lake inventory not generated explicitly for this study (E Glen 2022, personal communication, 22 July). 'Red standard value' and 'Green standard value' are calculated using our approach to calculating $A_d$, g, and $R_\infty$ (see Method section). The darker-coloured shading indicates the uncertainty of these values. 'Red range' and 'Green range' correspond to the possible depths achieved when upper and lower bounds are used for the parameter being varied. We note that in (c), the range and uncertainty of the depths are small and so appear as thin lines along the standard value lines.*"

In Figure 2, the parameters are varied together to find the uncertainty and range of the band-specific RTEs. We will add the following text to line 203 to clarify this, "*The uncertainties for each lake's band-specific RTE are calculated by co-varying all permutations of the RTE tuneable parameters.*"

**3) "More attention should have been paid to the selection of values of Ad in the RTE. Specifically, I would recommend the authors test a wider range of Ad values before this work is published. The author should read and refer to Dell et al. (2020), who use careful selection of Ad values to avoid the impact of slush."**

$A_d$ is intended to represent the reflectance of the lake bottom and we agree that it needs to be selected carefully. We have read Dell et al. (2020), to which we have added a citation on line 358, but note that their study focuses on Antarctic lakes, which often have more blurred boundaries, and tend to form on cleaner, flatter ice. For studies of lakes on Greenland it is reasonable to use the ring of pixels directly adjacent to the lake because:

1. Lakes tend to have well-defined boundaries (Figure 1 of the manuscript) and we have high confidence that we have detected the lake edge effectively.
2. Lakes occur in regions where cryoconite coverage is spatially variable. Pixels at the lake shoreline are therefore more likely to have more representative levels of cryoconite than distant pixels.
3. Lakes are surrounded by topographic highs – and so going too far away from the lake risks including $A_d$ values associated with dry snow.

We also note that in our analysis of the plausible ranges for each tuneable parameter, we test $A_d$ values in the range 0.1347–0.7724 (red) and 0.2055–0.7973 (green) which we consider to be comprehensive. Interestingly, this analysis reveals that the RTE is most sensitive to the uncertainty in the value of $A_d$ and we note that refining the method for determining $A_d$, especially at scale, should be a priority of future work.

We will add the following text to the revised manuscript at line 359, "*Dell et al. (2020) estimated $A_d$ from the sixth concentric ring of pixels around Antarctic lakes to reduce the potential impacts of slush on the RTE. In future work, methods of estimating $A_d$ on both Greenland and Antarctica should be tested due to the importance of $A_d$ in the RTE.*"

We will also add the Dell et al. (2020) citation to the reference list of the revised manuscript.

**4) "Despite your comments r.e. averaging the red and the green band depths, I would like to see some evidence for this method not being suitable moving forward, particularly given its use in both Pope et al. (2016) and Williamson et al. (2018). I understand what you are saying with regards to the plateauing effect caused using the red band, but it would be more convincing if you could provide evidence for this. You also do not mention the fact that previous studies have averaged these depths until the discussion, I would advise mentioning this much earlier on."**

As requested, we will now mention this much earlier on. Specifically, at line 95, we will add the following text, "*Although previous studies have averaged the depths retrieved from the red band RTE and the panchromatic band of Landsat 8 (Pope et al., 2016; Williamson et al., 2018), we do not do so within this study as Sentinel-2 does not have a panchromatic band. Additionally, this study specifically aims to understand the uncertainties associated with applying the physically based RTE to data acquired at a single band, and so an empirical averaging without a clear physical justification does not serve the purposes of this research.*"

At the end of section 2.2, we will also add the following text, "*We do not average band-specific depth estimates here, for the reasons outlined previously; however, we do note that this has been done in previous studies (e.g. Pope et al., 2016; Williamson et al., 2018).*"

We will also add/edit the following text at line 366, "*With a larger amount of ArcticDEM and/or ICESat-2 data, we suggest that future research could combine multiple satellite bands (Adegun et al., 2023) and data sources as inputs to a machine learning model and generate a well-constrained depth-*

*detection product using a data-driven approach, as opposed to the model-derived approach we use here.*"

We will also add the Adegun et al. (2023) citation to the reference list.

**5) "L74: Does this region only contain active lakes?"**

No, it also contains non-active lakes. We will alter line 74 to read, "*This region contains both active (repeatedly filling and draining) and non-active lakes…*"

**6) "L134-136: For the section starting 'we manually appraised', please can you better clarify what was done here, I can't make sense of it."**

To clarify the method, we will alter lines 131-135 to read, "*To estimate $R_\infty$ we averaged the reflectance of the ten darkest pixels in each substitute image, after manually filtering out pixels obviously associated with sediment traces or sensor-related scanning issues. This is slightly different to the way that $R_\infty$ has been calculated in previous studies but does not produce values that are appreciably different.*" in the revised manuscript.

**7) "L160: Whilst I understand that it is more likely for lake bathymetry to change over 11 months for Lake 5, surely the lake bathymetry could feasibly change over any lake and time period?"**

Although we agree with the reviewer that lake bathymetry could feasibly change over any lake and any period given the enhanced ablation one would expect at the lake bottom, the shape of the lake basin is dictated by the bedrock topography meaning that large changes through this mechanism are unlikely over the time periods in our study. To reflect this, we will remove the reference to Lake 5 in line 160. In the revised manuscript, the text on lines 159-162 will read, "*As the location and shape of supraglacial lakes are determined by bedrock topography (Echelmeyer et al., 1991), we assume there should be little change in the bathymetry of the lake basins between the data acquisition dates (see Sect. 3.1 for further details).*"

**8) "L165: For the paragraph containing line 165 (which details the method used to calculate each lake's depth from the DEM) and the two following paragraphs, much more detail is needed. It is very unclear how you carried out your methodology."**

We will add/edit the following text at line 163 to the end of section 2.3, "*In ArcticDEM, full lakes are represented by flat surfaces. To measure their depth, we need to examine the shape of the basin before it has filled or after it has drained. As drained lakes have similar characteristics to perpetually dry surface depressions, we had to first identify which depressions in the DEMs were associated with active lakes. To identify lakes that drain in our study region, we followed the approach outlined in Bowling et al. (2019). This takes all DEMs covering our study area in the ArcticDEM dataset and stacks them, then interrogates the variance of the stack, with areas of high standard deviation indicating potentially active lakes. We filter to identify pixels where the standard deviation lies in the range of 2-7 m; below this threshold, variation in elevation can arise from artefacts in the DEM; and ICESat-2 depth detection is limited to lakes up to 7 m deep (Fair et al., 2020). We then cross-*

*referenced these areas with the locations of known supraglacial lakes, and the availability of ICESat-2 data, to generate our sample (Appendix A.1).*

*We set the lake level in the empty DEM to be consistent with the ICESat-2 data, under the assumption that the ICESat-2 and ArcticDEM data are spatially coregistered; i.e. we identified the DEM elevation value at either end of the ICESat-2 transects where ICESat-2 depths are zero, averaged these values and subtracted the average from the entire DEM.*

*Due to the sparse temporal sampling of ArcticDEM, and the need to resolve empty basins, the DEMs are not temporally concurrent with the ICESat-2 and Sentinel-2 data. As a result, the smallest period between the ArcticDEM and ICESat-2 acquisition dates was approximately two months (Lake 4), and the largest period was approximately 11 months (Lake 5) (Table A1). As the location and shape of supraglacial lakes are determined by bedrock topography (Echelmeyer et al., 1991), we assume there should be little change in the bathymetry of the lake basins between the data acquisition dates (see Sect. 3.1 for further details)."*

**9) "L194: lake surface?"**

To clarify the manual delineation procedure for ICESat-2, we will alter lines 192-196 to read, "*We invited 10 altimetry experts to manually digitise the lake bathymetry from the refraction-corrected ATLAS ATL03 photon data plots using an online digitisation tool (https://apps.automeris.io/wpd/).*" This section was relevant only for a previous version of the ICESat-2 bathymetry data set and so this comment will not be relevant to the revised manuscript.

**10) "L269-L275: You only talk about the red depths in detail here, what about the green depths?"**

We will add the following text to the revised manuscript at line 275, "*The green band RTE shows a different pattern to that of the red band RTE. From the location of the cloud in relation to the XY line, we see that the green band RTE typically overestimates depth compared to ArcticDEM. The plateau depths of the green band RTE for these lakes are not visible but the size of the cloud gives an indication of the larger spread of values compared to the red band RTE.*"

Further discussion regarding the green band data in Figure 5 can be found in the Discussion section of the original manuscript at lines 317-323.

**11) "L325: I am not sure 'However' is the right word to use here."**

We propose altering the manuscript at lines 325-326 to read, "*However, the large variances in volume estimation between the green and red RTE depths and the ArcticDEM DEMs have contrasting implications to both this assertion and to one another.*" to clarify the subject and context of the sentence.

**12) "L332-224: This section could do with some re-wording to improve its clarity."**

We assume that the reviewer is referring to lines 331-334 and will alter the text accordingly to read, "*With a lower estimate of the water available within the ice dynamics system, our ability to predict ice*

*calving rates at marine-terminating glaciers is affected. Specifically, it could cause us to underestimate the contribution of meltwater to localised ice velocities, potentially understating the role of meltwater in ice calving caused by glacier velocity increases.*"

**13) "L357: The authors should also consider referencing Moussavi et al. (2020) here."**

We will cite Moussavi et al. (2020) on line 357.

**14) "Figure 1: Caption – where are the background images from? Sentinel-2? North Arrow is hard to see in main map and subset maps."**

In the caption for Figure 1 we will add the following text, "*The background images in panels (1)-(5) are the Sentinel-2 tiles detailed in Table A1.*" In the revised manuscript we will alter the colour of the northing arrows.

**15) "Figure 3: For the Y-axis find a way to space out the text for 'ArcticDEM' and 'ICESat-2'"**

In the revised manuscript, we will space out the text for 'ArcticDEM' and 'ICESat-2'.

**16) "Figure 4: Please add North labels and scale bars to these plots! You also need to state that you show the ICESat-2 transects on the depth plots."**

In the revised manuscript, we will add northing arrows and scale bars to the true colour imagery plots to give context to the figure. In the caption, we will alter the text to read, "*The true colour imagery is from Sentinel-2 (Table A1). ICESat-2 transects are shown in orange on the true colour imagery and the depth difference plots.*"

**17) "Appendix A Section 1: The authors should consider moving comments on the quality of ICESat2 into the main text."**

We agree that comments on the quality of ICESat-2 would help to develop the main text and will add the following text to line 185, "*After limiting the potential lake inventory by the availability of ArcticDEM (Appendix A.1) we considered the quality of the available ICESat-2 data, where the highest quality translates to the basins which can be most easily delineated from ICESat-2 photon refraction i.e. we can see both the lake surface and bed returns of the photons. In doing so, we limited our lake selection to the five study lakes.*"

**18) "Table A1: What date was the imagery downloaded by you? – Apply this comment elsewhere too."**

We will add columns to each of the data tables (Table A1 and Table A2) to reflect the dates when the data was downloaded.

References within this response

Adegun, A.A., Viriri, S. & Tapamo, JR.: Review of deep learning methods for remote sensing satellite images classification: experimental survey and comparative analysis, J. Big Data, 10, 93, doi:10.1186/s40537-023-00772-x, 2023.

Bowling, J.S., Livingstone, S.J., Sole, A.J. and Chu, W.: Distribution and dynamics of Greenland subglacial lakes, Nat. Commun., 10, 2810, doi:10.1038/s41467-019-10821-w, 2019.

Dell, R., Arnold, N., Willis, I., Banwell, A., Williamson, A., Pritchard, H. and Orr, A.: Lateral meltwater transfer across an Antarctic ice shelf, Cryosphere, 14, 2313-2330, doi:10.5194/tc-14-2313-2020, 2020.

Echelmeyer, K., Clarke, T. and Harrison, W.: Surficial glaciology of Jakobshavns Isbræ, West Greenland: Part I. Surface morphology, J. Glaciol., 37, 368-382, doi:10.3189/s0022143000005803, 1991.

Fair, Z., Flanner, M., Brunt, K., Fricker, H., Gardner, A.: Using ICESat-2 and Operation IceBridge altimetry for supraglacial lake depth retrievals, Cryosphere, 14, 4253-4263, doi:10.5194/tc-14-4253-2020, 2020.

Moussavi, M., Pope, A., Halberstadt, A., Trusel, L., Cioffi, L. and Abdalati, W.: Antarctic supraglacial lake detection using Landsat 8 and Sentinel-2 imagery: Towards continental generation of lake volumes, Remote Sens-Basel, 12, 134, doi:10.3390/rs12010134, 2020.

Pope, A., Scambos, T., Moussavi, M., Tedesco, M., Willis, M., Shean, D. and Grigsby, S.: Estimating supraglacial lake depth in West Greenland using Landsat 8 and comparison with other multispectral methods, Cryosphere, 10, 15-27, doi:10.5194/tc-10-15-2016, 2016.

Williamson, A., Banwell, A., Willis, I. and Arnold, N.: Dual-satellite (Sentinel-2 and Landsat 8) remote sensing of supraglacial lakes in Greenland, Cryosphere, 12, 3045-3065, doi:10.5194/tc-12-3045-2018, 2018.

---

## Author Response (AR2)

Point-by-point reply to the editor – Manuscript entitled: Evaluation of satellite methods for estimating supraglacial lake depth in southwest Greenland, authored: Melling et al.

Manuscript iteration: Revision post review

Editor: Joseph MacGregor

We thank the editor for his comments on, and suggested edits to, our paper. Please find each comment given below in bold type, with our response following in normal text. All line numbers refer to the original manuscript.

**1) "18: I don't see what the sentence starting with "Our analysis…" adds to the sentences that precede it. It has just been made clear RTE uncertainty affects lake depths. Suggest deleting."**

We agree and have removed the sentence starting on line 18 from the revised manuscript.

**2) "21: Simpler: "likelihood is poorly constrained""**

We have changed line 21 to read, *"…which could mean that hydrofracture likelihood is poorly constrained…"* in the revised manuscript.

**3) "23: Simpler: "cryoconite on lakebed reflectance""**

We have altered line 23 to read, "*cryoconite on lakebed reflectance*" in the revised manuscript.

**4) "24: This study is clearly "data-driven", so this wording is odd. Although the RTE method includes a model, it is not so different from many other remote sensing analyses that apply a simple model to derive a higher-level product. Given what is written in the Discussion, I suggest changing "data-driven" to "multi-sensor" or "multi-mission". Another term may be more appropriate."**

We agree with the editor and have changed line 24 to read, "*multi-sensor*" as opposed to "*data-driven*" in the revised manuscript.

**5) "Figure 3: Could the RMSD and correlation numbers in the boxes be bolded to be more visible?"**

In the revised manuscript, we have bolded the annotations of Figure 3.

**6) "Figure 4: Most of the plots are indistinct blue or red blobs for which it is nearly impossible to meaningfully distinguish the depth/depth-difference values. I strongly suggest using a discrete coloring scheme at 1 m intervals instead (also apply to color bar). There is a minor loss of precision relative to what the methods can do but much is gained for the reader."**

We agree with the editor and have altered Figure 4 accordingly.

**7) "Figure 5: Bigger a/b panel labels. Suggest discrete coloring scheme again at 250# interval but not as critical as for Figure 4."**

We have altered Figure 5 accordingly within the revised manuscript.

**8) "323: result"**

We are unsure as to what the editor is referring to in this case and would ask for further clarification should the editor find the comment still relevant.

**9) "340-344: The last two sentences just seem like a repeat of earlier in the paragraph and don't add new information. I suggest some sort of "vice versa" for the sentence currently starting with "Constrastingly" (awkward) in the lake volume underestimation case."**

We have changed lines 340-344 to read, "*Conversely, use of the red band RTE can lead to underestimations of 63 % of the lake volume compared to ArcticDEM, which would potentially yield contrasting implications for our understanding of ice sheet dynamics.*" in the revised manuscript.